# Effect of *Botrytis cinerea* Activity on Glycol Composition and Concentration in Wines

Eszter Antal [1], Miklós Kállay [2], Zsuzsanna Varga [2,*] and Diána Nyitrai-Sárdy [2]

1 Diagnosticum Ltd., Attila Str. 126., 1047 Budapest, Hungary; eszter@diagnosticum.hu
2 Institute of Viticulture and Enology, Hungarian University of Agriculture and Life Sciences (MATE), Ménesi Str. 45., 1118 Budapest, Hungary; kallay.miklos@uni-mate.hu (M.K.); nyitraine.sardy.diana.agnes@uni-mate.hu (D.N.-S.)
* Correspondence: varga.zsuzsanna@uni-mate.hu

**Abstract:** The content of 2,3-butanediol ((R,R) and meso isomers) and 1,2-propanediol in grape berries and "liquid samples" (all non-berry extracts) from the Tokaj wine region of Hungary was investigated. Our aim was to find out how the activity of *Botrytis cinerea* influences the concentrations of these compounds compared with healthy grapes. Based on the measured concentrations, we can make a distinction between healthy berries and noble, rotted, so-called aszú berries. We also investigated if there is a difference between finished aszú wines and liquids intended for aszú production. We wanted to investigate the amount and distribution of the stereoisomers of 2,3-butanediol and their proportions. The results of the HS-SPME-GC-FID analysis of the samples showed significant differences in the 2,3-butanediol content between healthy and botrytised, aszú berries and between liquid samples for aszú production and aszú wines. In the berry samples, meso-2,3-butanediol could not be detected, whereas in the liquid samples, we found good amounts of this isomer. This may be due to the fact that the appearance of the meso form of 2,3-butanediol is a consequence of alcoholic fermentation. Significant differences were found between wines from healthy grapes and wines from botrytised grapes in terms of the levo-2,3-butanediol content, so that from an analytical point of view, a difference can be made between wines from healthy and botrytised grapes. No significant differences were found between berry and liquid samples in terms of 1,2-propanediol concentrations during our tests.

**Keywords:** *Botrytis cinerea*; levo-2,3-butanediol; meso-2,3-butanediol; 1,2-propanediol

## 1. Introduction

Alcoholic fermentation produces a number of secondary products, and 2,3-butanediol is the second most abundant as a normal constituent of wine, making it an important potential source of aroma. Because of its very high threshold value (about 150 mg/L), 2,3-butanediol does not usually have a noticeable effect on the organoleptic properties of alcoholic beverages. In contrast, its amount in wine can considerably vary, with concentrations ranging from 0.2 to 3 g/L, averaging around 0.57 g/L. This high concentration can affect the wine's aroma due to its slightly bitter taste and the wine's body due to its viscosity [1]. During fermentation processes, measurable amounts of 1,2-propanediol (propylene glycol (1,2-PG)) are also formed. These compounds are virtually absent in unfermented must but are present in wines within certain limits [2].

1,2-Propanediol is well-known as a legal solvent in the food and beverage, pharmaceutical and cosmetic industries and is, therefore, found in many products of our daily lives. As a common solvent, it is used in the flavouring industry and can also be found in flavoured drinks and in flavoured drinks containing wine [3]. However, since wine cannot be flavoured per se [4], knowledge of its formation and presence in wine is also important from a legal point of view, especially with regard to the authenticity of wine. Previous results [5] have given naturally occurring concentrations between 10 and 30 mg/L in white

and red wines, but concentrations between 40 and 50 mg/L have also been described [5]. Some wines made with higher concentrations of crushed or enriched berry material showed elevated concentrations up to 140 mg/L. An important consideration in authenticity studies is the possible influence of wine fermentation conditions and yeast strains on the final concentration or enantiomeric ratio of 1,2-PG [5] and on malolactic fermentation (MLF). Taking into account the possibility of the natural production of 1,2-PG, some yeast species, considered as wild yeasts, such as *Hansenula*, *Candida*, *Pichia* or *Rhodoturola*, have been described to metabolise 1,2-PG under aerobic conditions in synthetic media. Su-zuki and Onishi described the ability of several different yeast genera and species to convert L-ramnose to 1,2-PG under aerobic conditions [6].

2,3-Butanediol is a well-known by-product of fermentation. A 2,3-butanediol content of between 0.4 and 1.0 g/L is an indicator of fermentation and shows whether ethyl alcohol and glycerol are the result of fermentation. It nominally occurs in four stereoisomeric forms (Figure 1), but two are identical mesoisomers, which form an easily crystallisable hydrate. The mesoisomer is optically inactive [7].

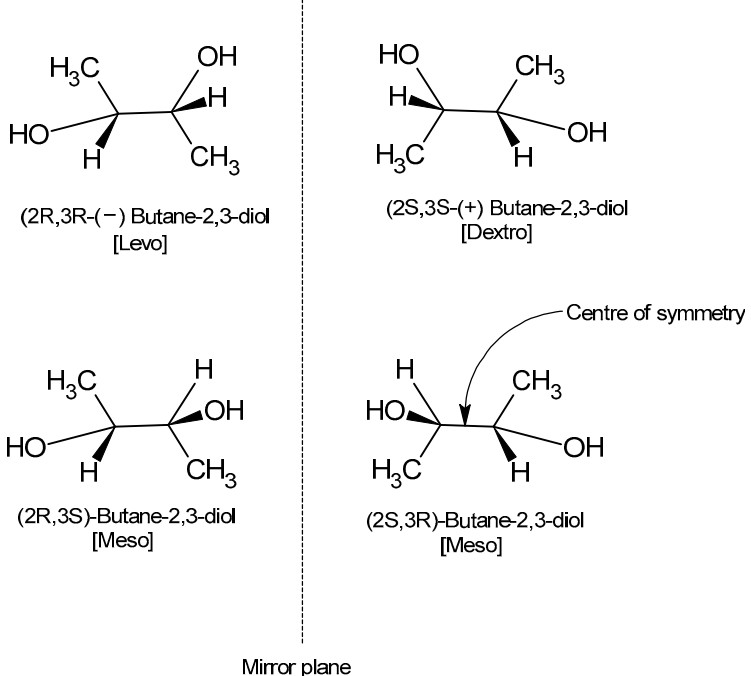

**Figure 1.** Stereoisomers of 2,3-butanediol.

Jung and co-workers [7] concluded that only the (R,R)- and meso isomers of 2,3-butanediol occur in wine, and that (S,S)-2,3-butanediol is not the true compound of wine. In addition, (R,R)-2,3-butanediol is the dominant isomer of 2,3-butanediol in wine. Its average value is 3.1 ± 0.7–1 g/L. The (R,R)/meso-butanediol ratio ranges from 1.6 to 5.2 g/L. It is considered that, in the future, the (R,R)/meso-2,3-butanediol ratio can be used to judge the authenticity of wines.

This seems to be contradicted by the study of Dankó and colleagues [8], who identified illiquid components in berries kept under conditions that induce nematode or bunch rot. In total, 30 volatile components were quantified. The Furmint berry samples affected by bunch rot also contained volatile organic compounds enriched in volatile matter [S,S]-2,3-butanediol. It was found that the progression of grey mould disease of furmint grapes was associated with increased volatile compound emissions, including [S,S]-2,3-butanediol.

The epidermal tissue, enzymatically loosened or destroyed by *Botrytis*, becomes permeable to must or surface saprobiont mycobacteria, so that other moulds, yeasts and acetic acid bacteria are also involved in the rotting process. Dankó et al. [8] observed that

the presence of acetic acid in must increases the formation of acetoin and its derivative 2,3-butanediol, possibly as a result of yeast activity.

The appearance of 2,3-butylene glycol in wine is, in fact, derived from acetoin, much of it being produced by yeasts during the fermentation of carbohydrates by the enzyme reduction of acetoin. Acetoin is produced by *Saccharomyces cerevisiae* in the early stages of fermentation, reaching a maximum about halfway through the fermentation process, and then rapidly decreases in the final stages of fol-yamat as it is reduced to 2,3-butylene glycol [9].

Several studies [8,10–12] have addressed the relationship between the mould *Botrytis cinerea* and 2,3-butanediol. *B. cinerea* is one of the most studied pen-fungi in viticulture because of its negative and positive effects. It is widespread and responsible for serious vineyard diseases that can damage yields and wine quality. Several studies have addressed the impact of *B. cinerea* on wine aroma, either in its harmful form or in its positive effects in causing noble rot [10].

The development of noble rot is considered as a beneficial process, resulting in the formation of bursting, chocolate-brown berries—aszú grains. These raisin-like berries are the main sources of flavour and aroma in botrytised sweet wines such as Hungarian aszú, French Sauternes and German Trockenbeerenauslese.

Hungary's most famous wine variety is Tokaji aszú. Its unique, delicate aroma and flavour are the result of a special wine-making technique involving the extraction of botrytised (aszú) grapes from noble rot, the extraction of the aszú grapes with new dry wine (sometimes with fermenting must) and a few years of ageing in small oak barrels.

Infection of grapes with *B. cinerea* leads to a change in the chemical composition of the grapes and associated wine, thus affecting the quality of the wine. The metabolic effect of *Botrytis* infection in Champagne base wine was investigated by Hong and colleagues [12]. They identified a number of components in grape must and wine, including 2,3-butanediol, and found that these components contributed to the metabolic differences between healthy and botrytised wines, both from an analytical and sensory point of view [12].

Although noble rot caused by *B. cinerea* is desirable for the production of good-quality sweet white wines, grey rot leads to undesirable or negative wine quality effects mainly through the degradation of aroma compounds and the production of off-flavours such as "mouldy" and "earthy" aromas.

Lower levels of 2,3-butanediol were found in the tested botrytised base wines than in healthy base wines [12].

Leskó and colleagues (2015) [13] investigated the amount of 2,3-butanediol and 1,2-propanediol produced after the fermentation of high sugar musts. They observed that higher initial sugar concentration promoted the formation of these fermentation by-products up to a certain point, but above the 250 g/L sugar content, the formation of these compounds was inhibited. Using a gas chromatography technique, they were also able to determine the stereoisomers of the resulting 2,3-butanediol [13].

Guymon and Crowell (1967) [14] investigated the levo and meso stereoisomers of 2,3-butanediol formed during the fermentation of grape juice (must) by gas chromatography. The effect of various parameters was investigated. It was found that the formation of 2,3-butanediol significantly increased with increasing fermentation temperature and with increasing initial sugar content of must [14].

The aim of this study is to investigate how the 2,3-butanediol content and isomer (levo/meso ratio) distribution of grape berries and liquid samples from the Tokaj-Hegyalja region of Hungary and the 1,2-propanediol content of grape berries and liquid samples accordingly evolve, whether it is a healthy berry or a berry neat rotten by *B. cinerea*, and whether it is a liquid of healthy grapes used for making aszú wine or aszú wine. Our question was if there is a significant difference in the amount of components tested according to whether the sample is botrytised or healthy.

## 2. Materials and Methods

### 2.1. Materials

All chemicals were analytical reagents. NaCl was purchased from Merck (Merck Life Science Kft., an affiliate of Merck KGaA, Darmstadt, Germany). Alcohol standard solutions of levo-2,3-butanediol, meso-2,3-butanediol, 1,2-propanediol and 1,3-propanediol (Internal standard) were from Merck. A stock solution (10 g/L of each compound) was prepared from the relative standard solutions and stored at 4 °C. Solutions at different concentrations were obtained by diluting stock standard solutions at the desired concentration.

### 2.2. Wine Sampling

Grape berry and "liquid samples" (everything that is not berry extract: wine, must) collected from the Tokaj wine region (Hungary) were analysed. The 62 different grape berries were healthy (33 berries) and botrytised (29 berries). Our liquid samples, 26 in total, were liquids for aszú production (10 types) and fermented aszú wines (16 types). The liquids for aszú production were must from healthy grape berries, partially fermented grape must, new wine in the process of fermentation and wine of the same vintage as the aszú berries; all of them are accepted in practice according to the "Product description of the Tokaj protected designation of origin", Version 9 [15]. The samples were provided by the producers, and, therefore, especially in the case of the berries, we accepted and did not overrule their judgement.

### 2.3. Preliminary Chemical Analysis on Wine Samples

The pH, total acidity (g/L as tartaric acid) and reducing sugar (g/L) were analysed according to the official methods established by the European Commission (EC) [16].

### 2.4. Sample Preparation

The berry samples had to be extracted for analytical tests. The extraction was performed with a 100 g berry sample. It was mixed with 100 mL of 10% $v/v$ ethanol [11]. After standing for half an hour, it was centrifuged and then filtered. The filtrate was collected in a 200 mL volumetric flask and then filled with 10% $v/v$ ethanol. Ethanol was chosen as the extraction agent because ethanol is also produced during alcoholic fermentation, and the concentration of ethanol determines the solubility of glycols in the final product. In summary, we have tried to model the extraction process that occurs during the winemaking process.

### 2.5. HS-SPME-GC-FID Analysis

The concentrations of 2,3-butanediol (levo- and meso-) and 1,2-propanediol were determined by gas chromatography (GC) using a flame ionisation detector (FID) and an automatic headspace (HS) solid-phase micro-extraction (SPME) procedure. For sample preparation, automated headspace–solid-phase micro-extraction (HS-SPME) analysis was used. About 10 mL of wine samples was pipetted and placed into a 20 mL headspace vial with 2 g of NaCl. Each sample was spiked with 500 μL of a solution of 1,3-propanediol (10 g/L in distilled water containing 15% $v/v$ of ethanol). The HS-SPME extraction procedure was carried out with a Combi PAL autosampler (from CTC Analytics AG, Zwingen, Switzerland) using an 85 μm CAR/PDMS fibre coating of 1 cm in length. Sample conditioning, extraction and headspace sampling were conducted using the agitator of the Combi PAL autosampler. The samples were incubated for 15 min at 85 °C followed by an extraction time of 3 min at 85 °C at a stirring rate of 400 rpm. Desorption was performed in the split/splitless injector of the gas chromatograph (GC) at a temperature of 200 °C. Desorption time was 5 min. Cleaning and conditioning of the fibre was conducted at 280 °C for 5 min under a nitrogen flow of 6 mL/min in the conditioning station of the autosampler before and after each analysis. Gas chromatographic analyses were conducted with a Shimadzu-2030 with FID detection. The separation column used was an HP 20M capillary column with 25.0 m × 0.20 mm i.d. and 0.20 μm film thickness. The GC was

used in splitless mode (splitless time in 2 min). The carrier gas was helium at a flow rate of 0.80 mL/min. The detector temperature was 250 °C. The GC oven was programmed as follows: the initial temperature was 50 °C (held for 2.40 min), then ramped at 13.3 °C/min to 200 °C and held for 12.05 min. The total run time was 25.73 min.

*2.6. Statistical Analysis*

Statistical evaluation of the measurement results was performed using the ANOVA program of Microsoft Excel (version 18.2106.12410.0, license: Microsoft Corporation, Redmond, Washington). The analytical results were evaluated by one-factor analysis of variance. The amounts of levo-2,3-butanediol (R,R-isomer), meso-2,3-butanediol and 1,2-propanediol were measured in berry samples and liquid samples of different qualities, and the evolution of concentrations is illustrated by boxplot diagrams. We compared whether there was a significant difference at the 95% ($p = 0.05$) probability level between the measured concentrations depending on whether the berries were healthy or infected with *B. cinerea* and whether the sample was a basic wine made from healthy grapes or an aszú.

**3. Results**

We tested mature, healthy and botrytised clusters from several vineyards in the Tokaj wine region, producing high-quality berries with noble rot. We also tested samples of grape juice from the same area at different stages of aszú wine production (liquid samples for soaking aszú berries). The mean values of the basic analysis and the ±standard deviation are given in Table 1.

**Table 1.** Basic analysis of the samples, mean and ±standard deviation of the measured values. Concentrations are given per unit berry weight (g/kg) for berry samples and per unit volume (g/L) for liquids.

| | | Reducing Sugar (g/kg), (g/L) | pH | Total Acidity (g/kg), (g/L) |
|---|---|---|---|---|
| Berry samples | Healthy | $208.8 \pm 21.7$ | $3.4 \pm 0.2$ | $5.8 \pm 0.3$ |
| | Noble rotten | $305.5 \pm 63.2$ | $3.3 \pm 0.2$ | $8.2 \pm 0.9$ |
| Liquid samples | Liquid for aszú Production | $15.7 \pm 37.4$ | $3.0 \pm 0.2$ | $7.0 \pm 0.9$ |
| | Aszú wine | $142.5 \pm 34.6$ | $3.1 \pm 0.2$ | $8.9 \pm 1.6$ |

In case of reducing the sugar content, the large variation between the results is due to the fact that there was a very large difference in the sugar content of the samples tested, both in terms of the sugar content of the berries at harvest and also among the liquid samples: there were some that were fermented completely dry and some that still had a low sugar content. In the course of the work, we did not require that the sugar contents were similar.

In the analysis of the 2,3-butanediol content of the berries, the meso-form could not be detected. Figure 2 shows the evolution of the levo-2,3-butanediol (R,R-isomer; hereafter referred to as R,R-isomer) content of berry samples at different stages. When tested by one-factor analysis of variance, it was found that there was a significant difference between the healthy grape berries and aszú berries at the 95% significance level ($p = 0.05$). Thus, the amount of levo-2,3-butanediol distinguishes these berry varieties from each other. The highest amount of le-vo-2,3-butanediol (1219.8 mg/kg) was measured in the aszú berries.

When analysing "liquid samples" (everything that is not berry extract: wine, must), we were able to measure both the levo and meso isomers of 2,3-butanediol. This leads to the primary conclusion that the appearance of the meso form of 2,3-butanediol is a consequence of alcoholic fermentation. The aszú wines contained both levo-2,3-butanediol (717.3 mg/L) and meso-2,3-butanediol (411.7 mg/L) in the highest concentrations.

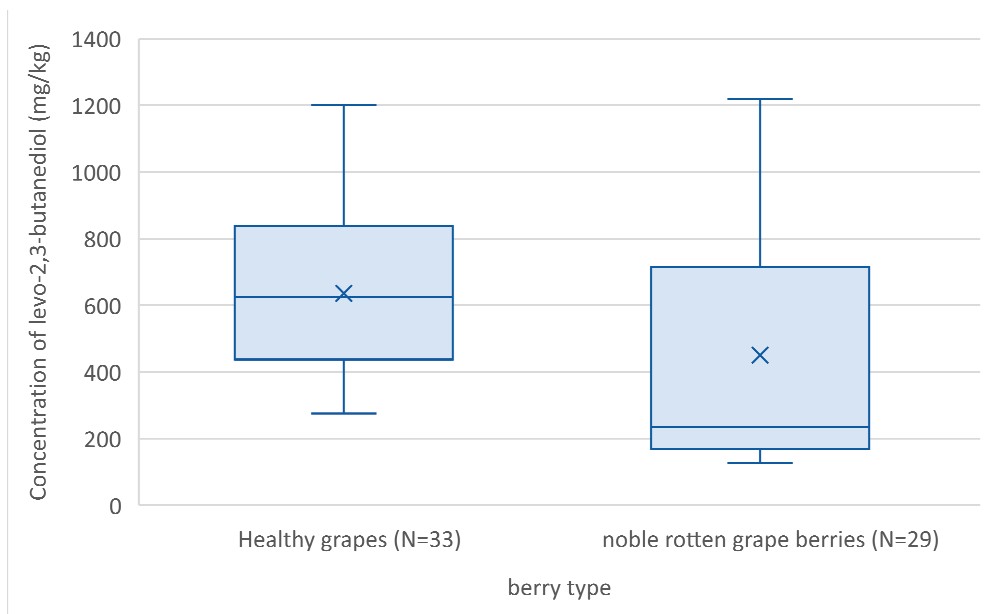

**Figure 2.** Boxplot of levo-2,3-butanediol concentrations of different types of berries.

The average levo- and meso-2,3-butanediol concentrations of the liquid and berry samples are shown in Figure 3. The amount of meso-2,3-butanediol was lower than that of levo-2,3-butanediol in both liquid types. In case of berry samples, however, the levo-2,3-butanediol content was higher in the healthy berries.

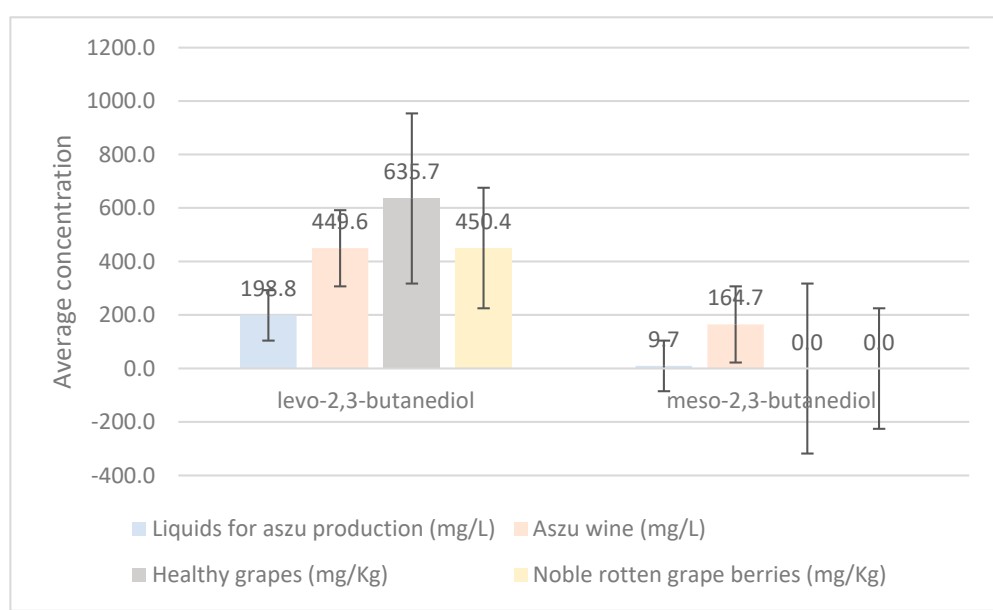

**Figure 3.** Comparison of average amounts of levo- and meso-2,3-butanediol measured in liquid and berry samples.

The results were analysed by one-factor analysis of variance and showed that at the 95% ($p = 0.05$) significance level, there was a difference between the aszú wines and the liquors for aszú production, both in terms of the amount of levo-2,3-butanediol and the amount of meso-2,3-butanediol.

We examined how the average amounts of levo-2,3-butanediol and meso-2,3-butanediol in the liquid samples relate to each other (Figure 4).

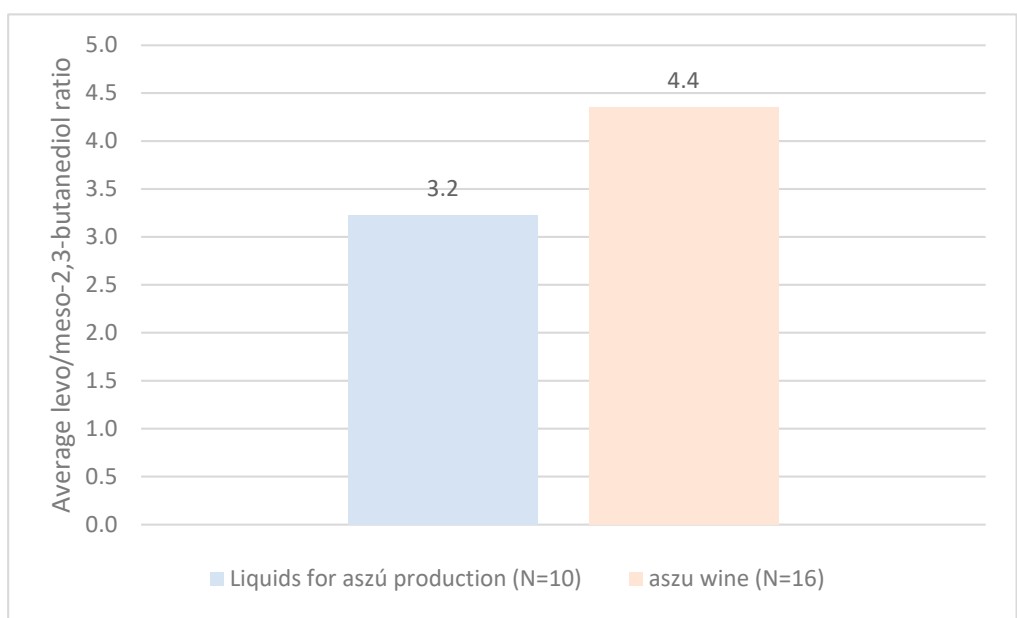

**Figure 4.** Ratio of average concentrations of levo- and meso-2,3-butanediol measured in liquid samples.

The trend of the levo/meso ratio shows that the ratio is higher for aszú wines (Figure 4).

Figures 5 and 6 show the evolution of 1,2-propanediol concentrations in the berry and liquid samples. The 1,2-propanediol content in berry samples varies within a wide range (healthy berries: 36.6–248.2 mg/kg, fortified berries: 21.6–510.8 mg/kg) (Figure 5), and the highest concentration was measured in the case of the aszú berries (510.8 mg/kg). By comparing the healthy berries and dried berries by one-factor analysis of variance, we found that there was no significant difference between them in terms of their 1,2-propanediol concentration at the 95% ($p = 0.05$) probability level.

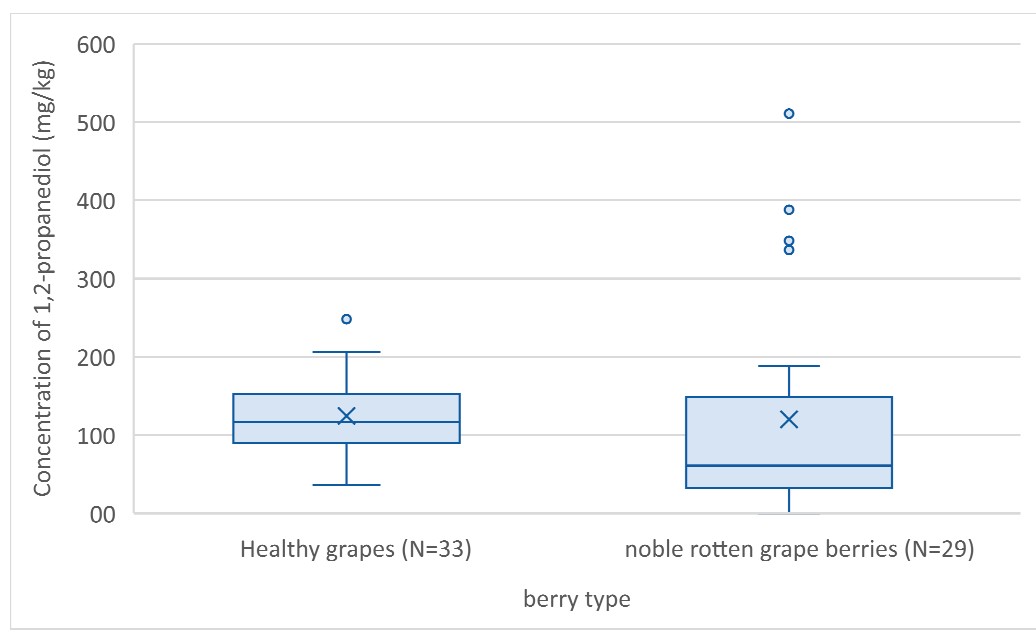

**Figure 5.** Boxplot of 1,2-propanediol concentrations of different types of berries (°outliers).

When looking at the concentrations of 1,2-propanediol in the liquids (Figure 6), it is observed that, apart from a few outliers (693.5; 394.4; 224.1; 103.1 mg/L), the concentrations measured in the samples remain below 100 mg/L. In case of liquids for aszú production, we

measured concentrations between 43.1 and 103.1 mg/L and between 30.9 and 693.5 mg/L in aszú wines, which is a very wide range.

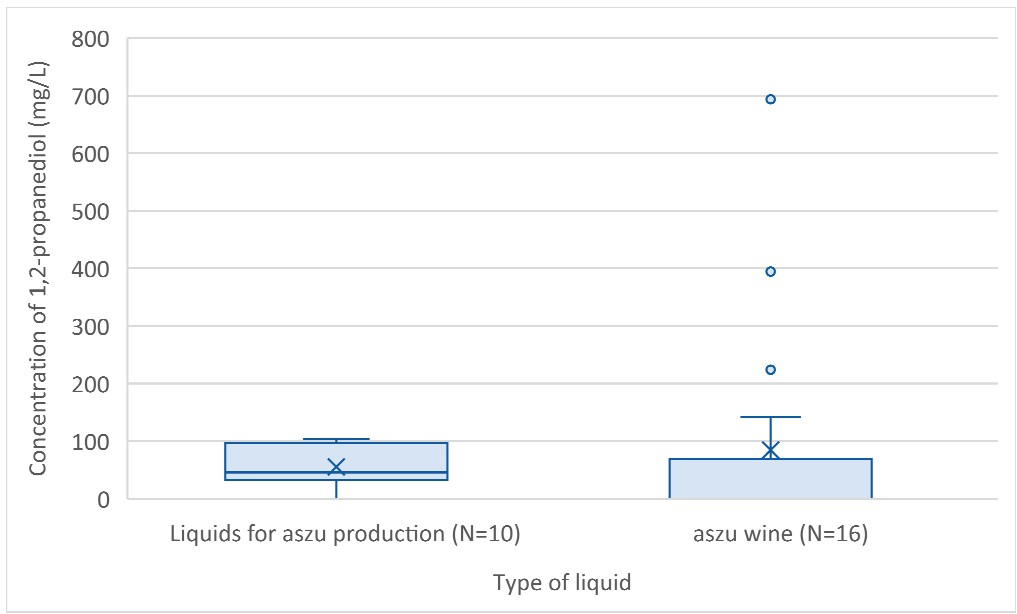

**Figure 6.** Boxplot of 1,2-propanediol concentrations of different types of liquid (°outliers).

There is no significant difference at the 95% ($p$ = 0.05) probability level between the 1,2-propanediol content of liquids intended for aszú production and aszú wines.

## 4. Discussion

The tests were carried out on grapes subjected to noble rot (botrytisation) and wines from these grapes. The concentrations of 2,3-butanediol isomers and 1,2-propanediol were compared with healthy grapes and wines from healthy grapes. The aim of the work was to find out how the amount of levo-2,3-butanediol (R,R-isomer) and meso-2,3-butanediol and 1,2-propanediol in the samples tested varied depending on whether the sample was botrytised or not.

First, we analysed the results of the berry samples and found that meso-2,3-butanediol could not be detected, whereas for the liquid samples, we found good measurable amounts of this isomer. The primary conclusion drawn was that the appearance of the meso form of 2,3-butanediol is a consequence of alcoholic fermentation.

The mean levo-2,3-butanediol content of the healthy berry samples was 635.7 mg/kg (N = 33), and the mean levo-2,3-butanediol content of the aszú berry samples was 397.7 mg/kg (N = 29). The average levo-2,3-butanediol concentration in the liquids for aszú was 198.8 mg/L (N = 10), and for aszú wines, it was 449.6 mg/L (N = 16) (Figure 3) [17]. The average levo-2,3-butanediol content of healthy berries was higher than that measured in the aszú berries, yet the aszú wines had the higher levo-2,3-butanediol content compared with the samples used for aszú production. This can be explained by the higher sugar content of the aszú berries compared with the sugar content of the healthy berries. Son and colleagues (2009) [17] found that the 2,3-butanediol level in wines may be related to the sugar level in grapes, with a positive correlation. In addition to this, infection of grapes with *B. cinerea* is always associated with the development of acetic acid bacteria on the berries [18,19]. These bacteria produce acetic acid, which during the fermentation process is converted into acetoin and then into 2,3-butanediol. This results in a higher 2,3-butanediol content in botrytised wines compared with healthy wines.

For both berry samples and liquid samples, results of the statistical analysis showed that there were significant differences for both isomers tested (levo- and meso-2,3-butanediol); thus, these isomers contribute to the differences between healthy and botrytised grapes

and their wines (healthy and botrytised wines) from an analytical point of view. Thus, the amounts of levo- and meso-2,3-butanediol distinguish the berry varieties and liquid samples tested. This result is consistent with the observations of Hong and colleagues (2011) [12].

We found that botrytised wines had higher concentrations of both isomers (levo- and meso-) than liquids intended for distillation. These liquids for distillation are musts from healthy grapes, partially fermented grape musts, new wines still in fermentation and wines. It was also observed that levo-isomer is produced in higher quantities than meso-isomer. This contradicts the finding of Hong and colleagues (2011) [12]. Hong and co-workers used 1H NMR techniques to investigate the amount of 2,3-butanediol in *Botrytis*-infected sparkling base wine and healthy base wine. They found lower 2,3-butanediol levels in botrytised base wines than in healthy base wines.

When looking at the levo/meso ratio in the liquids, the trend showed that this ratio is higher in aszú wines than in liquids intended for aszú production. In order to obtain a more accurate picture, it is necessary to investigate the levo- and meso-2,3-butanediol content of more wines from healthy grapes and aszú wines.

No significant differences were found between berry samples and liquid samples in terms of 1,2-propanediol concentrations in the samples tested.

**Author Contributions:** Conceptualisation, E.A. and M.K.; methodology, D.N.-S.; resources, E.A.; writing—original draft preparation, E.A. and M.K.; writing—review and editing, Z.V. and D.N.-S.; supervision, M.K. All authors have read and agreed to the published version of the manuscript.

**Funding:** This research received no external funding.

**Institutional Review Board Statement:** Not applicable.

**Informed Consent Statement:** Not applicable.

**Data Availability Statement:** The data presented in this study are available on request from the corresponding author.

**Conflicts of Interest:** The authors declare no conflict of interest.

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
