# Peer review of "Effect of Botrytis cinerea Activity on Glycol Composition and Concentration in Wines"

_fermentation, doi:10.3390/fermentation9050493_

Round 1

Reviewer 1 Report

The authors wanted to  investigate the amount and distribution of the stereoisomers of 2,3-butanediol 16 and their proportions in wines. However, there are still many major issues present in the manuscript.

1. Important research results should be presented in the abstract of the paper.

2. “given naturally occurring concentrations between 10 and 30 mg/L..” needs references.

3.The image is too large and needs to be merged.

Author Response

Thank you for your valuable review and helpful suggestions. Here are our responses:

  1. Important research results should be presented in the abstract of the paper.

Corrected.

  1. “given naturally occurring concentrations between 10 and 30 mg/L..” needs references.

References provided.

3.The image is too large and needs to be merged.

We need some more information. Exactly which images need merging. We gladly edit the images as suggested.

Reviewer 2 Report

Line 139: Grape berry and "liquid samples" (everything that is not berry extract) – this is not clear, need to be more accurate

Line 140: The 62 different grape berries were healthy and botrytised – present more details, like how many healthy and how many botrytised berries

Line 142: The liquids for aszú production were must, partially fermented grape 142 must, new wine in the process of fermentation and wine of the same vintage as the aszú 143 berries, - how many from each group?

Line 153: The berry samples had to be extracted for analytical tests. The extraction was performed with a 100 g berry sample. It was mixed with 100 mL of 10 v/v% ethanol. – why in ethanol? please explain, references needed

282-285 – references needed

Figure 3 and 4 – what is “Aszúáztatásra szánt folyadékminta” – I can not find the explanation in the text

the discussion part is very modest, For the quality of this manuscript work, it would be important if this part was more advanced.

Literature/references needs updating and amplification. There is definitely a lack of up-to-date literature data on the subject – need to be completed

Author Response

Thank you for your valuable review and your helpful suggestions. Here are our responses:

Line 139: Grape berry and "liquid samples" (everything that is not berry extract) – this is not clear, need to be more accurate

Corrected.

Line 140: The 62 different grape berries were healthy and botrytised – present more details, like how many healthy and how many botrytised berries

Corrected.

Line 142: The liquids for aszú production were must, partially fermented grape 142 must, new wine in the process of fermentation and wine of the same vintage as the aszú 143 berries, - how many from each group?

Corrected in the manuscript.

Line 153: The berry samples had to be extracted for analytical tests. The extraction was performed with a 100 g berry sample. It was mixed with 100 mL of 10 v/v% ethanol. – why in ethanol? please explain, references needed

References provided, and ethanol explained.

282-285 – references needed

References provided.

Figure 3 and 4 – what is “Aszúáztatásra szánt folyadékminta” – I can not find the explanation in the text

Corrected.

the discussion part is very modest, For the quality of this manuscript work, it would be important if this part was more advanced.

Discussion improved.

Literature/references needs updating and amplification. There is definitely a lack of up-to-date literature data on the subject – need to be completed

References have been updated.